# Magnetic Particle Imaging-Guided Thermal Simulations for Magnetic Particle Hyperthermia

**DOI:** 10.3390/nano14121059

**Published:** 2024-06-20

**Authors:** Hayden Carlton, Nageshwar Arepally, Sean Healy, Anirudh Sharma, Sarah Ptashnik, Maureen Schickel, Matt Newgren, Patrick Goodwill, Anilchandra Attaluri, Robert Ivkov

**Affiliations:** 1Department of Radiation Oncology and Molecular Radiation Sciences, The Johns Hopkins University School of Medicine, Baltimore, MD 21205, USA; hcarlto1@jhmi.edu (H.C.); shealy7@gatech.edu (S.H.); asharm55@jhmi.edu (A.S.); 2Department of Mechanical Engineering, School of Science, Engineering, and Technology, The Pennsylvania State University—Harrisburg, Middletown, PA 17057, USA; nageshwar25150@gmail.com (N.A.); aua473@psu.edu (A.A.); 3Materialise NV, 3001 Leuven, Belgium; sarah.ptashnik@materialise.com (S.P.); schickel.1@outlook.com (M.S.); 4Magnetic Insight Inc., Alameda, CA 94502, USA; mnewgren@magneticinsight.com (M.N.); goodwill@magneticinsight.com (P.G.); 5Department of Oncology, Sydney Kimmel Comprehensive Cancer Center, School of Medicine, Johns Hopkins University, Baltimore, MD 21205, USA; 6Department of Mechanical Engineering, Whiting School of Engineering, Johns Hopkins University, Baltimore, MD 21218, USA; 7Department of Materials Science and Engineering, Whiting School of Engineering, Johns Hopkins University, Baltimore, MD 21218, USA

**Keywords:** magnetic nanoparticles, magnetic particle imaging, magnetic hyperthermia, image guidance, finite element analysis

## Abstract

Magnetic particle hyperthermia (MPH) enables the direct heating of solid tumors with alternating magnetic fields (AMFs). One challenge with MPH is the unknown particle distribution in tissue after injection. Magnetic particle imaging (MPI) can measure the nanoparticle content and distribution in tissue after delivery. The objective of this study was to develop a clinically translatable protocol that incorporates MPI data into finite element calculations for simulating tissue temperatures during MPH. To verify the protocol, we conducted MPH experiments in tumor-bearing mouse cadavers. Five 8–10-week-old female BALB/c mice bearing subcutaneous 4T1 tumors were anesthetized and received intratumor injections of Synomag^®^-S90 nanoparticles. Immediately following injection, the mice were euthanized and imaged, and the tumors were heated with an AMF. We used the Mimics Innovation Suite to create a 3D mesh of the tumor from micro-computerized tomography data and spatial index MPI to generate a scaled heating function for the heat transfer calculations. The processed imaging data were incorporated into a finite element solver, COMSOL Multiphysics^®^. The upper and lower bounds of the simulated tumor temperatures for all five cadavers demonstrated agreement with the experimental temperature measurements, thus verifying the protocol. These results demonstrate the utility of MPI to guide predictive thermal calculations for MPH treatment planning.

## 1. Introduction

Magnetic particle hyperthermia (MPH) is an interstitial thermal therapy approved for treating recurrent glioblastoma with external beam radiation therapy [1]. First proposed in the mid-20th century, clinical MPH involves the intratumor delivery of magnetic nanoparticles (MNPs), which are heated via magnetic hysteresis loss by exposing the region to an alternating magnetic field (AMF) [2]. The magnetocaloric effect has garnered interest for applications in hyperthermia; however, it is prominent in doped manganite magnetic materials and not magnetite/maghemite [3]. For hyperthermia, treatment effectiveness requires the control of thermal energy to heat the tumor to a mild temperature (41–45 °C) for a prescribed time while minimizing the temperature rise in normal tissue. The duration of exposure at an elevated temperature (or “time-at-temperature”) defines the thermal dose [4], which is often expressed by the isoeffect dose metric cumulative equivalent minutes of exposure referenced against 43 °C (CEM43), close to the thermal breakpoint temperature (42.5 °C) of human cells [5,6]. Depending on the thermal dose (typically >15 min), hyperthermia can sensitize cells to radiation [7,8,9] or chemotherapy [10,11], increase blood perfusion to reduce tumor hypoxia [12,13,14], stimulate anti-tumor immune signaling [15,16,17], or it can be directly cytotoxic [18].

While MPH offers substantial advantages for intervention, technical challenges continue to inhibit wider clinical acceptance. Among these, an inability to accurately measure the MNP concentration and its distribution in tissues presents barriers to developing reliable MPH clinical workflows that provide adequate quality assurance measures to compare the delivered thermal doses with the initial prescriptive treatment plan [19]. As sources of heat, the nanoparticle content and distribution determine the thermal dose for MPH; therefore, knowing these in both tumor and surrounding tissues becomes essential when addressing the clinical requirements for reliability and quality in patient care. Previous efforts to quantify the nanoparticles’ distribution have relied on the analyses of computerized tomography (CT) scans [1,20], but there are some limitations. Foremost, the CT signal from the MNPs must be differentiated from the tissue signal; this can prove difficult for intratumor concentrations of less than ~10 g/L [1,19,21].

Magnetic particle imaging (MPI) [22] is a tracer imaging modality that measures time-varying responses of the magnetization vector of a sample and offers advantages for MPH over anatomical imaging. Early efforts demonstrated that the MPI signal correlated with the intratumor nanoparticle content, which can be used to predict a temperature rise when a region is exposed to AMFs [23]. These initial studies were extended to demonstrate how MPI can be used to monitor changes in MNP distribution after MPH [24]. Further advances integrated MPI with an AMF heater and used the MPI gradient field to restrict heating to a particular area within the AMF coil [25,26]. One group developed a computational model for predicting the spatio-thermal resolution of combined MPI/MPH systems [27]. A fully automated prototype AMF heating platform that enables spatially confined heating in a user-selected region of interest has also been described [28]. Buchholz et al. integrated MPI/MPH functionality into a platform that included MPI-based thermometry to monitor safety [29]; the same group also discussed proposed hyperthermia platforms that can integrate with commercial MPI scanners [30].

The technology exists to implement MPI-guided MPH; however, we must be able to incorporate MPI data into predictive thermal simulations to accurately estimate the intratumor temperature. Finite element (FEM) and finite difference methods (FDM) applications are available for other hyperthermia modalities. In those cases, the software enables the user to estimate the thermal dose within specified volumes to create thermal dose contours of the tumor and margins [31,32]. Treatment planning applications for hyperthermia include HyperPlan [33,34,35], SEMCAD X [36,37], and Plan2Heat [38,39]. In some cases, commercial finite element analysis (FEA) applications have been validated for hyperthermia [40]. We have previously validated COMSOL Multiphysics^®^ for preclinical MPH [41]. Lacking in those previous efforts was a direct knowledge of the MNP distribution in tissue or a tissue-mimicking phantom [42]. MNP distribution in tumors and surrounding tissue is subject to considerable individual variability, irrespective of the method used to deliver MNPs [9]. This fact is understudied and often underappreciated, yet it can be a singular source of uncertainty and unreliability in predictive computations of tissue temperature with MPH. It thus becomes essential to develop robust methods to ascertain the MNP tissue content and distribution in each tumor after delivery. Previous work demonstrated integrating MPI data into heat transfer simulations; however, approaches and experimental verification were limited [43,44]. Here, we describe the results of an effort to develop and verify a clinically translatable thermal simulation workflow that uses 3D MPI data as input for finite element computations to predict tumor temperature during a simulated MPH (Figure 1). To process the MPI and micro CT data, we used an imaging-data processing suite that has clinical utility. Verification was achieved by comparing temperatures measured from tumors in mouse cadavers that were heated by AMF-activated MNPs with predicted temperatures of simulated MPHs, where we used experimental temperature data as the initial and boundary conditions for the simulations.

## 2. Materials and Methods

### 2.1. Postmortem Animal Studies

#### 2.1.1. Mouse Models

All animal study procedures were conducted according to the protocol approved by the Johns Hopkins University Animal Care and Use Committee (JHU ACUC). The project/protocol identification number is not publicly available due to the JHU institutional policy and can be provided on request from the corresponding author. Five 8–10-week-old female BALB/c mice (Jackson Laboratory, Bar Harbor, ME, USA) were used. All mice were fed a normal diet and water ad libitum, maintained at a 12 h light/12 h dark cycle, and monitored daily for signs of distress or pain.

#### 2.1.2. Cell Line and Tumor Implantation

A vial of the 4T1 murine mammary carcinoma cell line [ER/PR/HER2 negative] was purchased from the American Type Culture Collection (ATCC, Manassas, VA, USA) and maintained according to the supplier’s recommendations. Roswell Park Memorial Institute (RPMI) 1640 media with 10% heat-inactivated fetal bovine serum (FBS) was used to grow the cells. For each mouse, approximately 50,000 cells were suspended in 100 µL of PBS and injected subcutaneously into the right thigh. Once the tumors were palpable, we measured them daily using calipers until they reached a volume between 100–500 mm^3^.

#### 2.1.3. MNPs

Hydroxyethyl starch-coated Synomag^®^-S90 iron oxide (mixture of magnetite and maghemite) nanoparticles (Lot #: 14422105–01; micromod Partikeltechnologie GmbH, Rostock, Germany) suspended in water and 50 mg of Fe/mL were used as received [45]. Photon correlation spectroscopy data from the manufacturer indicated a z-average diameter of 109.8 nm with a polydispersity index of 0.092. The particle concentration was verified using a Ferene-s assay [46]. In brief, we first digested the nanoparticles in an acetate buffer with ascorbic acid for at least 20 h. The iron concentration was determined by comparison with reference standards using UV/vis spectrophotometry. The heating rate of the MNPs was estimated using our previously published transient pulse analysis and was reported as specific loss power (SLP) [47]. A sample of nanoparticles at a concentration of 1 mg of Fe/mL H_2_O was heated at a peak AMF amplitude of 15 mT and at 50% duty (60 s ON/60 s OFF). Each pulse was analyzed and fitted to a non-adiabatic lumped mass heat transfer model, from which the SLP was calculated.

#### 2.1.4. Intratumor MNP Injections

For the MNP dose (i.e., MNP mass) calculation, we used a heat transfer approach to model the MNPs as a spherical uniform heat source [48,49,50] to identify the minimum concentration (*c*) needed to increase the temperature, Δ*T*, of a tumor having radius *R* (Equation (1)):(1)$$c=\frac{3∆Tk}{SLP\times R^{2}}.$$

Here, *k* is the isotropic coefficient of the thermal conductivity of the tumor. Using the measured SLP (496 ± 28 W/g) for the nanoparticle lot used (Appendix A) and assuming idealized conditions, we estimated the minimum MNP concentration required to raise the temperature of a tumor having radius *R* = 0.5 cm and *k* = 0.6 W/(m°C) from 37 °C to 43 °C (Equation (2)):(2)$$c=\frac{3\times 6℃\times 0.6\frac{W}{m℃}}{496\frac{W}{g}\times {\;\left(0.005m\right)}^{2}}=0.870\frac{mgFe}{mLTumor}.$$

Then, we adjusted the injection volume and concentration to achieve a target intratumor concentration of ~2 mg Fe/mL tumor to compensate for particle loss and heterogeneous intratumor distribution. Tumor size, for dose determination, was measured with calipers, and the MNP injected volume was adjusted to achieve 2 mg of Fe/mL tumor concentration (Table 1).

Mice were anesthetized via inhalation of 1–2% isoflurane mixed with O_2_ delivered through a nose cone. The nose cone remained in place for the duration of the MNP injection. MNPs were injected into each tumor at a rate of 2.5 µL/min using a syringe pump (Pump 11 Elite, Harvard Apparatus, Holliston, MA, USA), followed immediately by euthanasia. We decided to perform the experiments with mouse cadavers in order to remove the effects of temperature-dependent perfusion and tissue damage accumulation to better isolate the temperature change induced by the MNPs.

#### 2.1.5. MPI Scanner and Imaging

We used a Momentum^®^ MPI scanner (Magnetic Insight, Inc., Alameda, CA, USA) to generate 3D images of the MNPs within each tumor. The scanner was equipped with a custom mouse holder to limit lateral movement. We used three ~1 µL aliquots of Synomag^®^-S90 MNPs in microcentrifuge tubes secured to the sample holder within the scanning region as fiducial markers. Each isotropic scan was measured using the ‘Standard mode’ scanner configuration and an excitation field amplitude of 5 mT, with a gradient field of 5.7 T/m and a drive frequency of 45 kHz. Each 3D image consisted of 21 radial scans.

#### 2.1.6. The microCT

We used an IVIS SpectrumCT In Vivo Imaging System (Perkin-Elmer, Shelton, CT, USA) for the anatomical imaging, which was co-registered with the MPI scans. The scans (50 kV at 1 mA) consisted of 720 projections, each with an exposure time of 20 ms. We performed microCT immediately after MPI using the MPI sample holder. A 3D-printed adapter enabled the integration of the MPI holder into the IVIS scanner. We adjusted the scan area to include the mouse cadaver as well as the fiducials within the area of interest.

#### 2.1.7. AMF Heating

All tissue heating was performed on the HYPER device previously described [28]. The selected AMF amplitude for MNP heating by the solenoid was 15 mT ± 10% (12 kA/m) at a set frequency of 341.25 kHz, which is within the Hergt–Dutz biological limit of 5 × 10^9^ A/(m-s) [51]. We verified the amplitude with a 1D magnetic field probe (AMF LifeSystems, LLC, Auburn Hills, MI, USA). The temperature in the sample area encompassed by the solenoid was maintained at 37 ± 0.1 °C by a closed-loop water circulating system.

Before heating the tumors, mouse cadavers were individually placed into plastic bags, which were sealed and immersed into a circulating water bath (Polyscience, Niles, IL, USA) set at 37 °C for 15 min. Mouse cadavers were then removed from the water bath and plastic bags and then placed into a custom 3D-printed sample holder, also maintained at 37.0 ± 0.1 °C by circulated heated water. After securing the cadaver to the sample holder, a single probe was inserted into the tumor to the approximate geometric center. Temperatures were recorded at 1 s intervals. An additional probe was inserted into the rectum to monitor the core body temperature at similar intervals. Each tumor underwent two separate, consecutive heating trials. The first trial consisted of continuous heating at 15 mT for 30 min, or until the tumor exceeded 51 °C. After the trial, the tumor was allowed to cool until 37 °C. When the tumor temperature returned to 37 °C, it was heated at 15 mT at a 67% duty cycle (60 s ON/30 s OFF) for 20 cycles. We performed pulsed heating in addition to continuous heating to mimic the power modulation that may occur during MPH therapy.

### 2.2. Image Analysis and Computational Modeling

#### 2.2.1. Co-Registration

We used the Mimics Innovation Suite (MIS) Research v.25 (Materialise NV, Leuven, Belgium) with the FEA module for all image analyses, including segmentation, co-registration, conversion to 3D parts, and meshing. While we used the Research version of the software, MIS Medical has FDA 501k medical clearance. The suite consists of two main software: (1) Mimics v26.0^®^, which we used to create masks and 3D parts from the data, and (2) 3-matic^®^ v18.0 for smoothing and re-meshing. For the MPI-guided simulations, MPI and anatomical imaging data were co-registered to ensure alignment of the shared scale and coordinate system. Mathematically, co-registration transforms each voxel, or 3D pixel, at point *x* (*x*, *y*, *z*) to point *y* (*x*′, *y*′, *z*′), as shown in Equation (3):(3)$${\left[\begin{array}{cccc} a_{11} & a_{12} & a_{13} & a_{14}\\ a_{21} & a_{22} & a_{23} & a_{24}\\ a_{31} & a_{32} & a_{33} & a_{34}\\ 0 & 0 & 0 & 1 \end{array}\right]}_{A}\times {\left[\begin{array}{c} x\\ y\\ z\\ 1 \end{array}\right]}_{x}={\left[\begin{array}{c} x^{\prime }\\ y^{\prime }\\ z^{\prime }\\ 1 \end{array}\right]}_{y}.$$

Within matrix *A*, the top left 3 × 3 values scale or rotate the image, while the top right 3 × 1 values translate the image. Without this co-registration, the positions of the MNPs (heat generators) within the tumor would be unknown. Practically, we performed the co-registration using the Landmark Registration tool in Mimics^®^, where the locations of the fiducial markers in 3D space were matched in both the MPI and microCT image stacks (Figure 2). The application automatically generated a transformation matrix (matrix *A*), which we applied to the MPI data to spatially match the MNP locations within the tumor. The co-registered images for all 5 mice can be found in Appendix A.

#### 2.2.2. Imaging Data Calibration

To correlate the MPI signal within a single voxel to MNP thermal output, we prepared serial dilutions of the MNP suspension in 20 µL aliquots in plastic centrifuge tubes having concentrations 0, 0.25, 0.5, 2.5, 5.0, 25, and 50 mg of Fe/mL H_2_O. The tubes were placed onto the MPI sample holder to ensure separate and distinguishable samples. The calibration samples were then measured with the MPI using the same scanning parameters used to image the tumors. We then used Mimics^®^ to extract the maximum voxel intensity from each calibration sample. The MPI signal intensity of each voxel was mapped to an arbitrary grayscale by the Mimics^®^ software; we will refer to the units simply as a grayscale value (GV). Afterward, we plotted the maximum GV against the estimated volumetric thermal output from each sample (SLP × concentration) and used linear least squares regression to fit the data points (Figure 3). We assumed the SLP did not vary with the concentration or aggregation effects. A representative MPI scan and segmentations of the calibration samples can be found in Appendix A. The tabulated calibration curve can also be found in Appendix A.

#### 2.2.3. Mesh Generation

We created masks for both the tumor and MNP distribution using the built-in functions within Mimics^®^. The tumor boundary was segmented, and the created mask was then converted to a .STL file (3D part). We then imported the tumor-part file into 3-matic^®^ (Figure 4a), where the geometry surface was smoothed and re-meshed; a volume mesh was calculated (Figure 4b), and the final mesh was exported as a COMSOL^®^ mesh-part file (Figure 4c), using the built-in functions within 3-matic^®^. Similarly, the MNP distribution was segmented (Figure 4d), except that the distribution was converted directly to a voxel mesh in Mimics from the image data rather than being converted to a 3D part before meshing (Figure 4e). The MPI mask was created using a threshold segmentation, where the upper bound was the highest GV, and the lower bound was selected manually for each tumor, such that the majority of the signal within the tumor bounds was encompassed. Then, under material assignment, we used the calibration curve to convert the GV of each voxel to units of volumetric thermal output (W/m^3^), using the samples with the highest and lowest concentrations as references. A material property table was created, which contained the spatial coordinates of each voxel and the corresponding volumetric thermal energy generation (Figure 4f). A more detailed experimental procedure is described in the Appendix A.

#### 2.2.4. FEA Software and Mathematical Models

We used the COMSOL Multiphysics^®^ 6.2 (COMSOL, Burlington, MA, USA) advanced numerical methods software for all heat transfer calculations. The geometry of the computational model is illustrated schematically in Figure 5. The tumor mesh was modeled as a subcutaneous mass, where approximately 1/3 of the tumor volume was embedded within a cuboid geometry representing muscle. The computational phantom tumor and muscle resided within a cylindrical AMF coil, having dimensions similar to that of the experimental coil on the HYPER device.

We used two physics modules available in COMSOL, Heat Transfer in Solids and Magnetic Fields, as well as the Electromagnetic Heating multiphysics module to simulate eddy current heating. The material properties used for the simulation are summarized in Table 2. The Heat Transfer in Solids module implemented the Fourier conduction equation (Equation (4)),
(4)$$\rho c_{p}\frac{𝜕T}{𝜕t}=k\nabla^{2}T+Q_{MNP}+Q_{eddy}$$
where *ρ* is the density, *c_p_* is the specific heat at constant pressure, *k* is the coefficient of thermal conductivity, *T* is the temperature, *t* is the time, and *Q_MNP_* and *Q_eddy_* are the volumetric thermal power outputs from the hysteresis heating of the MNPs and Joule heating from induced eddy currents, respectively. For *Q_MNP_*, we imported the material properties table created in Mimics^®^ from the MPI data as a 3D interpolation function, which was used as a spatially variant heat source.

*Q_eddy_* was calculated by coupling the frequency domain of Maxwell’s equations (Equations (4)–(7)) with the conduction equation (Equation (4)),
(5)∇×H=J,
(6)B=∇×A,
(7)J=σE+jωD,
(8)E=−jωA,
where *J* is the current density, *H* is the magnetic field amplitude, *B* is the magnetic flux density, *A* is the magnetic vector potential, *σ* is the electrical conductivity, *E* is the electric field amplitude, *ω* is the angular frequency, and *D* is the electric flux density. As an initial condition, *A* was assumed zero in all dimensions, and Ampere’s law was applied to the phantom muscle and tumor geometries. The AMF coil was modeled as free space with a magnetic field of amplitude 12 kA/m, applied along the *z*-axis (along the coil length). Using the equation for *Q_eddy_* (Equation (9)), COMSOL solved the Joule heating contribution as follows: (9)$$Q_{eddy}=0.5Re\left(J\cdot E\right)+0.5Re\left(i\omega B\cdot H\right).$$

On the top surface of the tumor and muscle, a convective boundary condition (Equation (10)) was implemented using Newton’s law of cooling, with a constant ambient temperature (*T_∞_*) of 37 °C and a heat transfer coefficient (*h*) of 20 W/(m^2^·K).
(10)$$q=h\left(T-T_{\infty }\right),$$
where *q* is the convective heat flux. 

A uniform temperature boundary condition was used on the remaining five rectangular faces of the phantom muscle, where the temperature value was interpolated from the experimental cadaver rectal temperature at a given time point. The rectal temperature data can be seen in Appendix A. Additionally, we used experimental data for our initial conditions, where the initial temperature of the phantom tumor(s) and muscle geometries corresponded with the respective initial experimental temperatures measured, i.e., at *t* = 0. A default physics-based “fine” mesh was selected in COMSOL. We performed a frequency-transient study at 341.25 kHz for each phantom tumor to mirror the experimental heating. Specifically, simulated heating was conducted as one continuous heating trial and one pulsed heating trial (60 s ON/30 s OFF). Predicted temperatures from tumor phantoms were reported as a volumetric maximum, minimum, and average of the entire tumor geometry.

### 2.3. Uncertainty Quantitation

We used the Type A uncertainty evaluation, where valid statistical methods were used to treat our data, as defined by the National Institutes of Standards and Technology [53]. The MPI calibration was performed in triplicate, where the values used to convert the MPI GV to a volumetric thermal output consisted of the average calculated from the 3 replicate measurements. All statistical testing, linear regression, and analyses were performed using Prism 6 software (GraphPad, Boston, MA, USA).

## 3. Results 

All simulated tumor temperature values showed general agreement with the measured intratumor temperatures (Figure 6). For tumors 1, 3, and 4, the calculated average tumor phantom temperatures were ±3 °C of the experimental temperatures measured at the tumor center. Deviations between the simulated and experimental temperatures were observed with tumors 2 and 5, with the latter showing the largest discrepancy. In this case, most of the injected MNPs were concentrated on the left tumor boundary, while the remainder of the tumor (notably the tumor center and location of the probe) was virtually MNP-free. However, the experimental temperature plot for tumor 5 matches the minimum simulated temperature reasonably well, supporting the accuracy of the simulation. We observed significant variances of intratumor MNP and temperature distributions among the tumors after single-point injection, consistent with the previous observations from other tumor models (Figure 6) [9]. 

## 4. Discussion

### 4.1. MNP Distribution and Thermal Probe Placement

The results demonstrate the challenges to achieving a uniform distribution of MNPs within any tumor (constant concentration of MNPs throughout), even with percutaneous delivery, thus highlighting the importance of accounting for the MNP content and distribution in tissues prior to activating them with an AMF. One strategy used to mitigate the effects of unknown and heterogeneous MNP distributions was by manual or automated temperature control mechanisms [42]; however, their success relies on the appropriate placement of the thermal probe within the tumor. Consider tumor 5 (Figure 6). Based solely on the experimental data, we observed temperatures in the range of 40–50 °C for the duration of testing, which is ideal for MPH. Yet, based on the MPI data, the largest concentration of MNPs was along the tumor periphery, away from the position of the probe at the tumor center, resulting in the predicted temperatures being >twofold what was measured experimentally. The MPI data imply that such conditions in a live subject could have produced significant ablative tissue damage at the tumor boundary, assuming a single thermal probe placed in the tumor center as a reference. During treatment, such a disparate probe placement from the concentration of MNPs provides little useful information for accurate thermal dosimetry in the tumor and, worse, fails to anticipate the damage to surrounding normal tissues. MPI, with its enhanced sensitivity to changes in MNP concentration, provides the necessary critical information to locate such potential “hot spot” regions and avoid damaging healthy tissue when compared to works that use anatomical imaging alone.

### 4.2. Implications for Treatment Planning

Our aim in this work was to develop and verify the capability of MPI to guide thermal simulations using clinically relevant software. To achieve this, we conducted the computational analysis after experimental heating trials, where experimental temperature data was used to provide both boundary and initial conditions for simulations. If simulations are to be used for MPH treatment planning, initial and boundary conditions would be required as initial inputs prior to treatments commencing. Our results here show that reasonable approximations, e.g., subject temperature, could suffice, provided the simulations are used to estimate average, maximum, and minimum temperatures in the target volume (tumor). In the context of treatment planning, these values could be used to generate thermal dose contours to guide AMF power management to ensure treatments achieve an outcome within a prescribed thermal dose range. Certainly, the accuracy of correspondence between planned and achieved treatments would improve if temperature probe placement is performed with imaging guidance. MPI-guided simulations thus can provide the basis of MPH quality control to the researcher or clinician who is able to follow a pre-determined, in silico treatment plan. Additional investigation is needed to ascertain the limits of MPI-guided simulations for treatment planning.

### 4.3. Limitations

For combined MPI/MPH treatment, the MNPs are both therapeutic and tracer imaging agents; therefore, the selected nanoparticles must generate adequate heat with acceptable MPI tracer quality. For this study, we use commercially available magnetic nanoflowers, Synomag^®^, that exhibit suitable properties for both MPI and MPH [54]; however, the effect of the MNP dose on MPI image quality bears consideration. The MPI signal varies linearly with the MNP spatial density, i.e., concentration. A high-intensity signal generally produces quality images, but if the concentration is too high, the signal saturates the MPI detector, resulting in image aberrations. Corrective actions can be implemented, such as lowering the excitation field amplitude and increasing the gradient field magnitude, but relatively high MNP concentrations (>1 mg Fe/mL) associated with MPH therapies will likely produce image artifacts when compared to tumor models injected with MNPs at tracer concentrations (~µg Fe/mL). 

MPH requires accurate active thermometry to ensure that the prescribed thermal dose is achieved. Further, in vivo thermometry with MPH typically uses invasive probes (usually one) implanted directly within the tumor to measure the intratumor temperature. From a heat transfer perspective, single-point thermometry introduces severe limitations, especially for larger tumors, as seen with tumor 5 (Figure 6). To draw useful conclusions from a single temperature probe, one must assume the tumor is a lumped mass with a uniform spatial temperature distribution, which is only true for sufficiently small tumors that are uniformly loaded with MNPs. Future development of multi-point or volumetric thermometry would enhance the accuracy of MPI-guided simulations of MPH and reduce the associated uncertainties.

## 5. Conclusions

We developed and verified an MPI-guided thermal simulation workflow for use in MPH preclinical experiments. We conclude that our methodology successfully integrated MPI data into FEA software for thermal simulations. Verification of the proposed methodology was achieved by comparing the simulation results with corresponding experimentally heated tumor-bearing mouse cadavers that had been previously injected with MNPs. The general agreement between the experimental and simulated results raises confidence that this approach can overcome many of the technological limitations encountered with clinical MPH. Achieving further in vivo preclinical validation of this methodology will establish its potential for use in human patients. 

## Figures and Tables

**Figure 1 nanomaterials-14-01059-f001:**
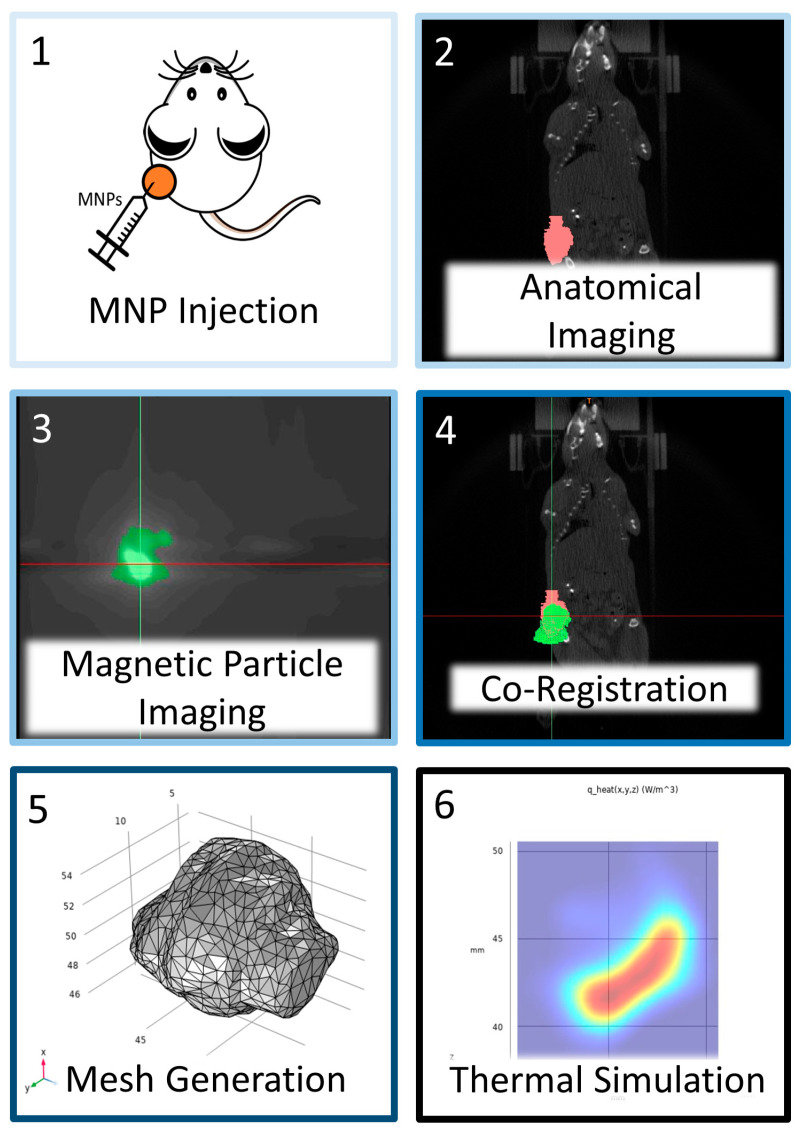
Schematic representation of workflow for magnetic particle imaging (MPI)-guided thermal simulations. Magnetic nanoparticles (MNPs) at a pre-determined dose were injected intratumorally. After MNP injection, the tumor was imaged with both MPI and microCT for anatomical reference. Using fiducial markers, the MPI data were co-registered with anatomical images to establish a common datum. The tumor geometry was segmented, converted to a 3D part, and volume-meshed and imported into FEA software (COMSOL Multiphysics 6.2); additionally, voxel intensities obtained from the MPI scan were converted into volumetric thermal output values from a calibration curve developed using measured MNP heating data. With the MPI values spatially registered and calibrated for heat output, the FEA simulation was performed.

**Figure 2 nanomaterials-14-01059-f002:**
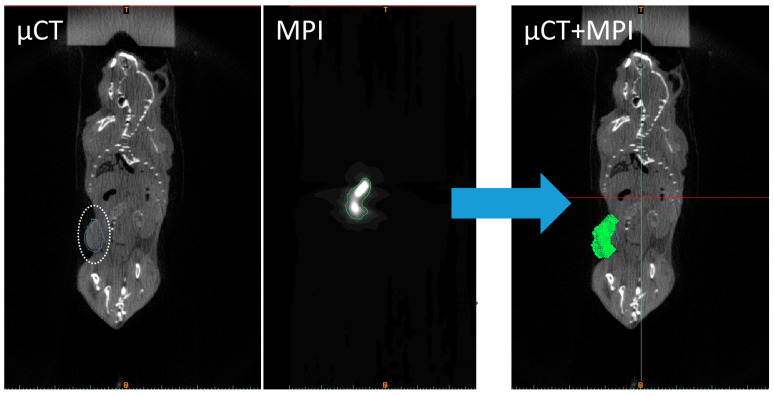
Co-registration is necessary to correlate MPI voxel intensity values with 3D coordinates within the segmented tumor (Mouse 1 is shown as an example). The micro(μ)CT and MPI of tumored mice after injection are shown, where the tumor is circled in white. The micro(μ)CT and MPI scans were co-registered within Mimics, and a transformation matrix was created. The transformation matrix obtained for each tumor was applied to the MPI scan and overlaid onto the micro(μ)CT scan.

**Figure 3 nanomaterials-14-01059-f003:**
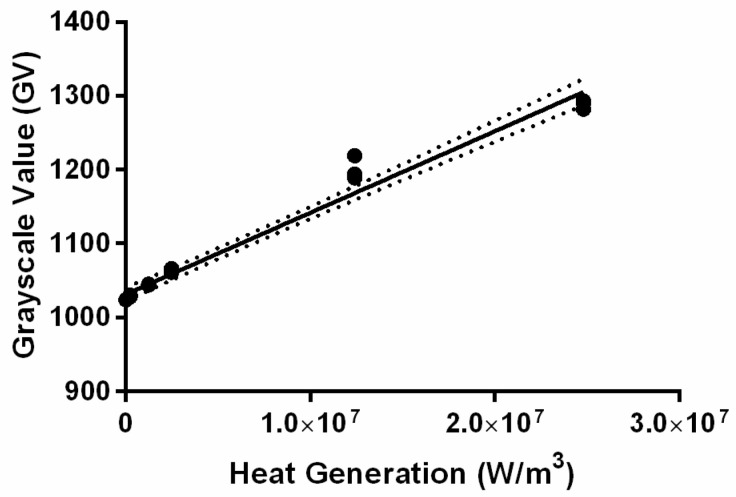
MNP calibration is essential to generate accurate MPI-guided simulations. The max grayscale and thermal generation values (calculated from SLP) were fitted to a linear equation using least squares regression to yield *y*-intercept of 1031 ± 4 GV and slope = (1.10 ± 0.04) × 10^−5^ W·m^−3^·GV^−1^ with R^2^ = 0.9746. Uncertainty is represented by 95% confidence bounds, which are shown as dashed lines in the figure.

**Figure 4 nanomaterials-14-01059-f004:**
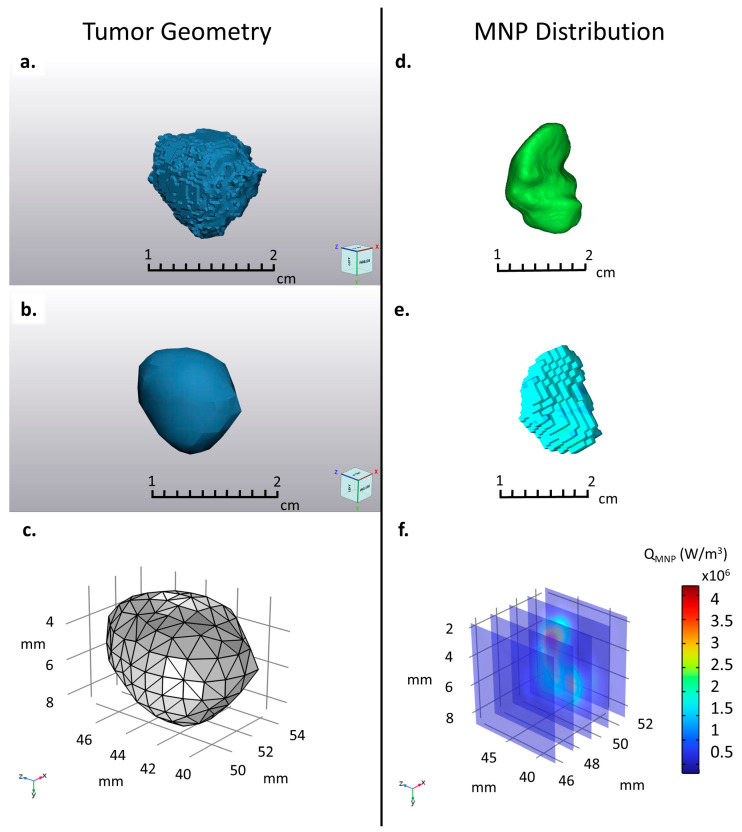
After co-registration and segmentation, the represented MNP distribution and tumor geometry (Mouse 3 shown as an example) were prepared for COMSOL. (**a**) After converting the tumor mask to a .stl file, it was imported into 3-matic, where it was (**b**) smoothed, re-meshed, and (**c**) converted to a COMSOL mesh file. (**d**) The MPI scan was segmented, and the resulting mask was used to represent the MNP distribution in the computational tumor phantom. (**e**) The voxel intensities were mapped onto the values of the MPI calibration curve using the Mimics^®^ material assignment tab. (**f**) The calibrated voxel values were then imported into COMSOL as a 3-argument interpolation function.

**Figure 5 nanomaterials-14-01059-f005:**
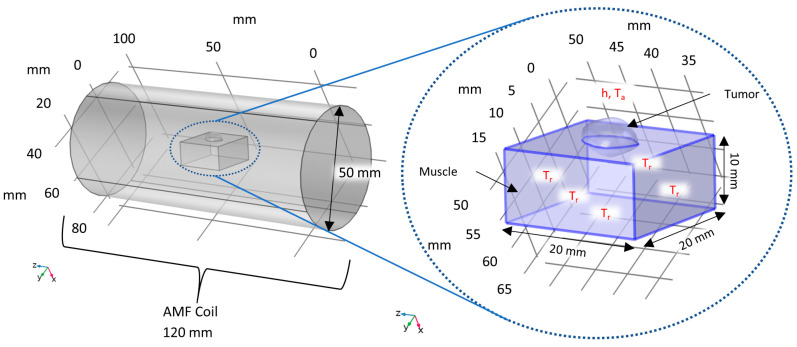
Schematic diagram of computational tumor and muscle phantom used to simulate MPH with COMSOL. The imported tumor geometry was embedded in a phantom body represented by a cuboid having properties of muscle, suspended in a phantom AMF coil with a diameter similar to that of the coil in the HYPER device. Constant temperature boundary conditions were assumed, having values obtained from experimental rectal temperatures, on the bottom and sides of the phantom body. We imposed a convective boundary condition on the top of the phantom tumor and muscle, with an assumed heat transfer coefficient and ambient temperature.

**Figure 6 nanomaterials-14-01059-f006:**
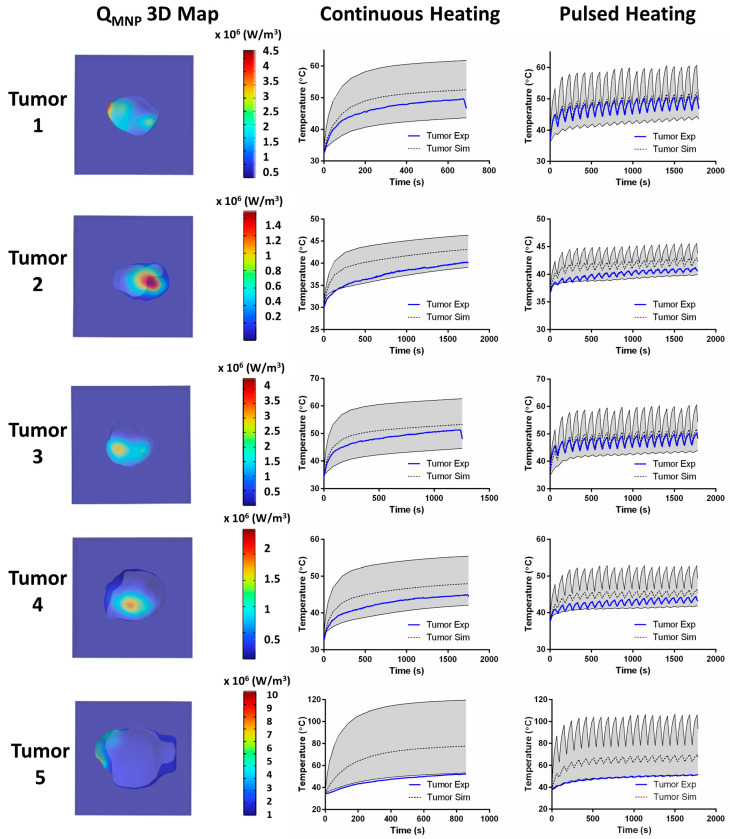
MPI-guided simulations of magnetic hyperthermia accurately predict intratumor temperatures measured experimentally in the center of murine 4T1 tumors. The left column shows the distribution of MNPs within the tumor from the MPI data that were used as inputs for simulated MPH in computational phantoms. The second column shows the results of the continuous heating trials, where the gray region encompasses the maximum and minimum predicted temperature values obtained from the computational phantoms, the dashed lines show the average simulation temperature, and the solid blue lines are the experimentally measured temperatures from the 4T1 tumor heating trials in mouse cadavers. The third column shows similar results, which were obtained from the pulsed heating trials.

**Table 1 nanomaterials-14-01059-t001:** Tumor sizes were measured with calipers and associated MNP injection volume.

Sample	Caliper Measured Tumor Volume (mm^3^)	MNP Injection Volume (µL) @ 50 mg Fe/mL
Tumor 1	144	5.6
Tumor 2	193	7.7
Tumor 3	280	11.2
Tumor 4	355	14.2
Tumor 5	455	18.2

**Table 2 nanomaterials-14-01059-t002:** Material properties used for COMSOL electromagnetic heating simulations.

Property	Muscle	Tumor	Ref.
Specific Heat at Constant Pressure (cp)	3421 J/(kg-K)	3760 J/(kg-K)	[43]
Density (ρ)	1090 kg/m^3^	1045 kg/m^3^	[43]
Thermal Conductivity (k)	0.49 W/(m-K)	0.51 W/(m-K)	[43]
Relative Permeability (µ_r_)	1	1	
Relative Permittivity (ε_r_)	2000	2000	[52]
Electrical Conductivity (σ)	0.23 S/m	0.23 S/m	[52]

## Data Availability

Data are publicly available on the Johns Hopkins Data Repository.

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
