# Peer review of "Magnetic Particle Imaging-Guided Thermal Simulations for Magnetic Particle Hyperthermia"

_nanomaterials, 2024, doi:10.3390/nano14121059_

Round 1

Reviewer 1 Report

Comments and Suggestions for Authors

Magnetic hyperthermia is a new, already used in practice, method of treating cancer in hard-to-treat areas by introducing biocompatible non-toxic magnetic nanoparticles into the tumor, their subsequent heating by an external electromagnetic field, leading to an increase in the temperature of tumor cells to +42°C and their subsequent death. 

Traditional methods of cancer tumor treatment are chemotherapy and radiation therapy, but the attention of scientists and physicians is turned to the search for alternative methods that would be less dangerous to the patient's health. Thus, the method of hyperthermia is considered as an alternative to the treatment of the last stages of cancer or as a supplement to traditional treatment, in which individual organs or parts of the organ affected by the pathological process are exposed to high temperature.  Unlike healthy tissue, the affected tissue has large intercellular spaces, high density of blood vessels and poorly developed lymph nodes. Heat exposure to such tissue leads to damage and further cell death (apoptosis), while healthy cells remain unaffected. The therapeutic effect of hyperthermia is limited to the temperature range from +38ºC to 46ºC. At a temperature of +38 ºC active proliferation of tumor cells is observed, at +39 ºC viability decreases, and at temperatures above +43 ºC their death is observed. Hyperthermia can also be used to reduce tumor size to operable states 

In the currently used hyperthermia methods (general hyperthermia, local hyperthermia using radiofrequency and microwave radiation as well as ultrasound), the heating of the tumor also causes a significant increase in the temperature of adjacent healthy tissues. Possible undesirable consequences of hyperthermia treatment are overheating, blood clots, burns and cardiovascular disorders. To control the thermal regime, the temperature is measured using sensors inserted into the treatment area. Thus, the main disadvantages of currently used hyperthermia techniques are low selectivity of exposure, as well as invasive method of temperature control.

Therefore, therapeutic techniques that utilize some form of localized exposure are considered preferable. The use of local hyperthermia in combination with radiation and/or chemotherapy improves the results of treatment of cancer patients by 20 - 40%. Currently, magnetofluidic hyperthermia is actively used for the treatment of carcinoma, glioblastoma and prostate cancer. There are two main ways of administering magnetic nanoparticles - direct injection of particles into the tumor and intravenous injection, which, in turn, is divided into two delivery methods: active and passive 

In this sense, the task of controlling the distribution of nanoparticles in the tumor and, consequently, the temperature distribution as a result of heating seems to be very relevant.

The authors present an original methodology of such a protocol, which can undoubtedly be in demand in real clinical practice.

The work is written so thoroughly and qualitatively that there are no significant questions to the authors. The paper belongs to the type of high quality papers when the authors preface the reviewer's questions.

I can congratulate the authors on this success.

My questions/comments are advisory in nature.

I think some clarifications could have made the paper simply perfect.

Why were the studies done on mouse cadavers? Probably more value could have been derived from in vivo results

In the description of heating mechanisms (line 43 page 2), the authors should also add magnetocaloric heating (it does not appear in the case of magnetite, but gives a significant contribution in the case of many other materials)

There (line 50-51) the authors very appropriately refer to recent work that emphasizes the possibility of simultaneous therapeutic effects of magnetic fields and radiation therapy 

https://pubs.aip.org/aip/jap/article-abstract/129/3/033902/347247/Promising-magnetic-nanoradiosensitizers-for?redirectedFrom=fulltext

Have Synomag nanoparticles been studied in any way? I mean the size, shape of the particles? Their size distribution? Line 138 page 4.

Was the Brezovic criterion taken into account in any way when selecting the parameters of the alternating magnetic field line 191 page 5

Are the results applicable to other cancers and other types of nanoparticles?

I suggest finalizing the article with these recommendations in mind

Author Response

Reviewer 1

  1. The work is written so thoroughly and qualitatively that there are no significant questions to the authors. The paper belongs to the type of high quality papers when the authors preface the reviewer's questions. I can congratulate the authors on this success.

Response: We appreciate the reviewer’s kind words!

  1. Why were the studies done on mouse cadavers? Probably more value could have been derived from in vivo results.

Response: Our decision to use cadavers rather than live mice was deliberate and carefully considered. Live mice actively regulate their physiological temperature in response to environmental temperature changes, or changing thermal loads/stresses. This dynamic and non-linear response shared by all vertebrates includes vasodilation/vasoconstriction, increased/decreased respiration, and perspiration (only in some mammals, not mice). It also presents numerous technical challenges for both simulations and experiments. We would need to account for dynamically changing boundary conditions in our simulations by considering Arrhenius perfusion and tissue damage accumulation in order to properly model the mouse in silico. By heating mouse cadavers, we simplified our simulations to ensure robust verification of the MNP thermal output, which was our main goal. Certainly, validation of our methodology will require testing in vivo, in live tumor-bearing animal models. Having verified our simulations-based approach in cadavers, we are able to focus our future validation studies on attributing thermoregulatory response to variances between predictions and experimental results. This reduces our problem to a single (though admittedly complex) variable. We have added a sentence discussing this in the Materials and Methods section.

  1. In the description of heating mechanisms (line 43 page 2), the authors should also add magnetocaloric heating (it does not appear in the case of magnetite, but gives a significant contribution in the case of many other materials).

Response: We thank the reviewer for the suggestion. We have included a sentence and reference to bring this to reader’s attention.

  1. There (line 50-51) the authors very appropriately refer to recent work that emphasizes the possibility of simultaneous therapeutic effects of magnetic fields and radiation therapy. https://pubs.aip.org/aip/jap/article-abstract/129/3/033902/347247/Promising-magnetic-nanoradiosensitizers-for?redirectedFrom=fulltext.

Response: We thank the reviewer for the suggestion; however, our comments regarding hyperthermia and radiosensitization in the manuscript refer to the general and well-established effect of thermal stress. The radiation sensitizing effects of hyperthermia to which we refer arise from both biological and physiological mechanisms, and are independent of hyperthermia modality. Thus, the effect(s) do not depend on either magnetic fields or on nanoparticles. The literature cited by the reviewer points to another phenomenon, i.e. radiation enhancement that occurs by stimulation of Auger or thermal electrons when high density and high z materials are irradiated by x-rays (depends on wavelength). This is an altogether different phenomenon from the one we describe in the manuscript, and is one that does not generate substantial (clinically relevant) effects with magnetic iron oxide nanoparticles.

  1. Have Synomag nanoparticles been studied in any way? I mean the size, shape of the particles? Their size distribution? Line 138 page 4.

Response: Yes, several published articles describe Synomag nanoparticles. We have included a new reference, which describes results of characterizations of the material properties of the Synomag nanoparticles. We have also included data values from the manufacturer: particle size and polydispersity index.

  1. Was the Brezovic criterion taken into account in any way when selecting the parameters of the alternating magnetic field line 191 page 5.

Response: We took into account the more relaxed Hergt-Dutz criterion of 5 ´ 109 A/m-s. We used a 12 kA/m field at 341.25 kHz, for a product of 4.095 ´ 109 A/m-s. We have included a sentence describing this criteria.

  1. Are the results applicable to other cancers and other types of nanoparticles?

Response: Yes. While we used a breast cancer model for this study, the methodology can be extended for other types of cancer. As long as the nanoparticles are magnetic, can generate heat within the tumor and produce a signal on the MPI with a reasonable spatial resolution, then this method can be used to simulate a thermal treatment.

Reviewer 2 Report

Comments and Suggestions for Authors

The manuscript developed a protocol that incorporates magnetic particle imaging (MPI) data into finite element calculations for simulating tissue temperature during magnetic particle hyperthermia (MPH). The results should be interesting for the future development of MPH treatment planning tools. I can recommend accepting of this manuscript after minor revisions.  

1) It’s suggested to avoid the abbreviation “MPI” in the title.

2) In the abstract, line 28, the abbreviation “CT” need to be defined.

3)  What does the “CEM43” (in line 50) mean?

Author Response

Reviewer 2

  1. It’s suggested to avoid the abbreviation “MPI” in the title.

Response: We appreciate the feedback! The acronym “MPI” has been changed to “Magnetic Particle Imaging”.

  1. In the abstract, line 28, the abbreviation “CT” need to be defined.

Response: Thank you for the comment! We have defined the acronym CT in the abstract.

  1. What does the “CEM43” (in line 50) mean?

Response: CEM43 is cumulative equivalent minutes (CEM) at a reference temperature of 43°C. It is an isoeffect dose parameter used in thermal medicine to measure thermal dose. We have revised the relevant section to clarify for readers.

Reviewer 3 Report

Comments and Suggestions for Authors

The objective of this study is to develop a protocol that incorporates MPI data into finite element calculations for simulating tissue temperature during MPH. To verify the protocol, the authorsconducted MPH experiments in tumor-bearing mouse cadavers. Although several experiments were performed, i suggest some revisions to improve the quality of the manuscript.

Although reported in previous studies, I suggest providing some characterizations of the designed materials, specifically morphology and physicochemical properties if any of MNPs.

Modify schematic rather than Data in Figure 1.

In the introduction, i suggest adding more studies on the reported literature

Better to seperate discussions from results.

Suggest challenges related to design or bioefficacy in the conclusion section.

Author Response

Reviewer 3

  1. Although reported in previous studies, I suggest providing some characterizations of the designed materials, specifically morphology and physicochemical properties if any of MNPs.

Response: Thank you for the recommendation. We have included a reference in the updated draft, which describes the characterization of Synomag nanoflower particles in detail. We also included the z-average particle diameter and polydispersity index provided by the manufacturer. For this study, we focused characterization on the relevant parameter, heating output or SLP.

  1. Modify schematic rather than Data in Figure 1.

Response: We thank the reviewer for the comment. We have modified Fig 1 and caption according to reviewer and editor suggestions to make clear to readers this is a schematic representing the workflow we developed and described in the manuscript.

  1. In the introduction, i suggest adding more studies on the reported literature.

Response: We have included additional references.

  1. Better to seperate discussions from results.

Response: We have separated the Results from the Discussion.

  1. Suggest challenges related to design or bioefficacy in the conclusion section.

Response: We appreciate the reviewer’s comment. We have included limitations of the study as its on subsection within the Discussion, and we have modified the Conclusions as suggested by editor.
